



# Gas-Equilibrium Membrane Inlet Mass Spectrometery (GE-MIMS) for water at high pressure

Matthias S. Brennwald[1], Antonio P. Rinaldi[2], Jocelyn Gisiger[3], Alba Zappone[2], and Rolf Kipfer[1,4]

[1]Eawag, Swiss Federal Institute of Aquatic Science and Technology, Dübendorf, Department of Water Resources and Drinking Water
[2]Swiss Federal Institute of Technology in Zurich, Department of Earth Sciences, Institute of Geophysics
[3]Solexperts AG, Switzerland
[4]Swiss Federal Institute of Technology in Zurich, Department of Earth Sciences, Institute of Geochemistry and Petrology

**Correspondence:** Matthias S. Brennwald (matthias.brennwald@eawag.ch)





**Abstract.**

Gas species are widely used as natural or artificial tracers to study fluid dynamics in environmental and geological systems. The recently developed gas-equilibrium membrane-inlet mass spectrometry (GE-MIMS) method is most useful for accurate and autonomous on-site quantification of dissolved gases in aquatic systems. GE-MIMS works by pumping water through a gas equilibrator module containing a gas headspace, which is separated from the water by a gas-permeable membrane. The partial pressures of the gas species in the headspace equilibrate with the gas concentrations in the water according to Henrys

Law, and are quantified with a mass spectrometer optimized for low gas consumption (miniRUEDI or similar). However, the fragile membrane structures of the commonly used equilibrator modules break down at water pressures $\gtrsim 3$ bar. These modules are therefore not suitable for use in deep geological systems or other environments with high water pressures. To this end, the SysMoG® MD membrane module (Solexperts AG, Switzerland; "SOMM") was developed to withstand water pressures of up to 100 bar. Compared to the conventionally used GE-MIMS equilibrator modules, the mechanically robust construction

of the SOMM module entails slow and potentially incomplete gas/water equilibration. We tested the gas-equilibration efficiency of the SOMM and developed an adapted protocol that allows correct operation of the SOMM for GE-MIMS analysis at high water pressures. This adapted SOMM GE-MIMS technique exhibits a very low gas consumption from the SOMM to maintain the gas/water equilibrium according to Henrys Law and provides the same analytical accuracy and precision as the conventional GE-MIMS technique. The analytical potential of the adapted SOMM GE-MIMS technique was demonstrated

in a high-pressure fluid-migration experiment in an underground rock laboratory. The new technique overcomes the pressure limitations of conventional gas equilibrators and thereby opens new opportunities for efficient and autonmous on-site quantification of dissolved gases in high-pressure environments, such as in research and monitoring of underground storage of $CO_2$ and waste deposits, or in the exploration of natural resources.



## 1    Introduction

Gas species are widely used as natural or artificial tracers to study fluid dynamics in environmental and geological systems. The recent development of mobile mass spectrometers for on-site gas analysis has paved the way for efficient on-site gas analysis and monitoring of fluids in environmental systems (Brennwald et al., 2016, 2020; Cassar et al., 2009; Chatton et al., 2017; Gentz and Schlüter, 2012; Kotiaho, 1996; Mächler et al., 2012; Manning et al., 2016; Schlüter and Gentz, 2008; Schmidt et al., 2015; Sommer et al., 2015; Visser et al., 2013). In particular, the gas-equilibrium membrane-inlet mass spectrometry (GE-

MIMS) method used with the miniRUEDI mass spectrometer system (Brennwald et al., 2016; Mächler et al., 2012) provides simple and practical means for accurate quantification of dissolved gas species (e.g., He, Ne, Ar, Kr, Xe, $H_2$, $N_2$, $O_2$, $CO_2$, $CH_4$, $C_3H_8$) in lakes, oceans, and groundwaters.

The GE-MIMS method works by pumping water through a gas equilibrator module containing a gas headspace, which is separated from the water by a gas-permeable membrane. The partial pressures of the gas species in the headspace equilibrate

with the gas concentrations in the water according to Henrys Law. The partial pressures of the gases are quantified by "sniffing" the gas headspace with the miniRUEDI mass spectrometer without disturbing the gas/water equilibrium. Unlike other membrane-inlet methods that rely on the dynamic gas flux across the membrane, the GE-MIMS method operates at gas/water equilibrium and is therefore insensitive to the membrane-specific gas-transfer dynamics. Hence, the GE-MIMS method allows accurate quantification of dissolved gas concentrations without the cumbersome and notoriously difficult calibration of the

dynamic gas-transfer characteristics of the membrane (Brennwald et al., 2016; Mächler et al., 2012).

The combination of the GE-MIMS method with the portable and robust miniRUEDI mass spectrometer allows autonomous on-site quantification of individual gas species during fieldwork at remote locations. The miniRUEDI is therefore widely used with GE-MIMS in environmental and geological research to study gas/water exchange processes, biogeochemical turnover, and the origin and transport of water and other fluids (Batlle-Aguilar et al., 2017; Berndt et al., 2016; Brennwald et al., 2022;

Giroud et al., 2023; Knapp et al., 2019; Lightfoot et al., 2022a, b; Mächler et al., 2013a, b; Moeck et al., 2017, 2021; Popp et al., 2019, 2020; Roques et al., 2020; Schilling et al., 2021; Tomonaga et al., 2019; Tyroller et al., 2018; Weber et al., 2018).

However, the GE-MIMS method is currently limited to applications with water pressures of approximately 3 bar or lower. This limitation is due to the mechanical design of the membrane modules (Liqui-Cel MM 1.7×5.5; "LIMM") that have been tested and validated for use with GE-MIMS. These membrane modules are optimized for efficient gas transfer across the mem-

brane, allowing rapid gas/water equilibration of the gas headspace with the water flowing through the membrane module. The fragile membrane structures used in these modules break down and are irreversibly damaged at high water pressures. Custom-designed pressure-reducing systems have been tested to lower the pressure of the water without affecting its gas concentrations


(Weber et al., 2021; Zappone et al., 2021). Howeber, such systems are expensive and their operation is challenging and involves a large and poorly constrained dilution of the sample water. The dissolved-gas measurements determined using such
pressure-reducing systems therefore tend to exhibit undesired data gaps and and large analytical uncertainties. The application of the GE-MIMS method has therefore remained challenging in systems with high water pressures, such as deep geological systems explored for natural resources or for underground storage of waste or $CO_2$.

The recently developed SysMoG® MD membrane module (patent pending, Solexperts AG, Switzerland; "SOMM") was designed to withstand water pressures of up to 100 bar. This module might, therefore, allow efficient application of the GE-
MIMS method in systems with high water pressures. Owing to its mechanically robust membrane construction, the SOMM module is expected to show much slower and potentially incomplete gas/water equilibration, which would fundamentally affect the correct functioning of the GE-MIMS method (similar as with small dead-end membrane probes as tested by Marion (2022) and Engelhardt (2023)).

Here, we present systematic tests of the gas-equilibration efficiency of the SOMM and its potential utility for GE-MIMS.
Based on these tests, we developed and validated an adapted GE-MIMS protocol for the accurate quantification of aqueous gas concentrations using the SOMM. This adapted SOMM GE-MIMS technique was implemented in a fluid-migration experiment at the Mont Terri Underground Rock Laboratory (Switzerland) to demonstrate its analytical potential for dissolved-gas analysis in high-pressure systems.

## 2 Materials and Methods

### 2.1 Gas equilibrator modules

The Liqui-Cel LIMM membrane module commonly used for GE-MIMS applications is constructed from a plastic shell (4.3 cm external diameter, 13 cm long) and contains an array of approximately 5000 parallel hollow fibers made of gas-permeable microporous polypropylene (0.3 mm external diameter, 120 mm effective length). The water flow passes through the lumen side of these hollow-fiber membrane elements, and dissolved gases are exchanged with the shell side gas volume. Depending on
the gas species considered, gas/water equilibrium is attained within 5-20 min (Mächler, 2012). The fragile membrane elements and the plastic shell can withstand water pressures of approximately 3 bar, but will break down at higher water pressures.

The Solexperts SOMM membrane module (patent pending) uses a stainless-steel shell (6 cm external diameter, 20 cm long) and contains six membrane elements, which are constructed using polymer membrane tubes (12.7 mm external diameter, 86 mm effective length). The water flow passes through the steel shell, and dissolved gases are exchanged with the gas phase in

the lumen of the membrane elements. The membrane tubes are internally supported by porous stainless-steel frits to prevent the

tubes from collapsing at high water pressures. The gas volumes of the membrane elements are connected in series by stainless

steel tubing to allow purging of the gas through the membrane elements and an external loop for gas sampling.

The dimensions relevant to the GE-MIMS operation with the LIMM and the SOMM are compared in Tab. 1. Due to the

mechanically robust construction of the SOMM, its membrane thickness ($W$) and the gas-volume to membrane-area ratio

($V/A$) are of a magnitude higher than those of the LIMM. Additionally, the non-porous SOMM membrane material is expected

to be less permeable to gases than the microporous membrane material of the LIMM (Luis, 2018). As a result, the gas/water

equilibration in the SOMM will be substantially slower than in the LIMM, and may even be incomplete if gases are consumed

rapidly from the headspace for analysis with the miniRUEDI.

[Table 1 about here.]

**2.2   Test setup**

Fig. 1 shows the test setup used to assess the gas/water exchange dynamics in the SOMM. The SOMM was compared to the

LIMM, which serves as a reference for accurate quantification of dissolved gases using the GE-MIMS method (Brennwald

et al., 2016; Popp et al., 2019). Both membrane modules were fed with the same groundwater pumped from a local aquifer at a

flow rate of 1 L/min, and their gas headspaces were connected to separate gas inlets of a miniRUEDI instrument. The He, Ar

and $N_2$ concentrations the groundwater are known to be constant, whereas the $O_2$ concentration may change slightly over time

(see also Sec. 3.2).

The gas in the lumen of the SOMM membrane elements and the dead volumes of the connecting tubes was actively purged

through an external circulation loop (approx. 2 cm³ internal volume) using a small membrane pump (type FF 20 KTDC-M,

KNF Switzerland). In contrast, the LIMM setup does not require purging of the shell side gas, which is connected to the

miniRUEDI gas inlet without any dead volumes.

[Figure 1 about here.]

**2.3   Experiment I: Gas-exchange time scale and gas depletion due to miniRUEDI gas consumption**

In a first experiment, we determined the time scale for gas/water exchange in the SOMM and the degree of gas depletion due

to gas consumption by the miniRUEDI. To this end, we filled the gas volume of the SOMM with pure He gas at 1 bar. The He

was transferred across the membrane and removed by the water flow. At the same time, gas species dissolved in the water were

transferred into the SOMM gas volume. The dynamic evolution of the gas species in the in the gas volume was monitored with



the miniRUEDI until steady-state conditions were attained. The miniRUEDI was set to continuous sampling, whereby the inlet connection was switched between the two membrane modules at 6.5 min intervals. The relative time fraction ("duty cycle") spent for gas sampling from a given module was therefore 50 % for both the LIMM and the SOMM.

## 2.4 Experiment II: Modified protocol for unbiased GE-MIMS analysis

In a second experiment, we adapted the GE-MIMS protocol according to the SOMM gas-exchange dynamics determined in Experiment I. We decreased the duty cycle of the miniRUEDI gas sampling from the SOMM module with the aim of attaining solubility equilibrium between the water and the gas volume in the SOMM membrane elements. We tested and evaluated the accuracy and precision of this adapted SOMM GE-MIMS technique by comparison with the standard GE-MIMS technique

using a LIMM for reference.

# 3 Test results and GE-MIMS adaptions

## 3.1 Experiment I

Fig. 2 shows the evolution of the He, $N_2$, $O_2$ and Ar partial pressures observed in the SOMM gas volume during Experiment I. The He spike is slowly removed from the SOMM gas volume, whereby the He partial pressure followed an exponential curve.

At the same time, $N_2$, $O_2$ and Ar accumulated in the the SOMM gas volume. The partial pressures of all gases evolved following first-order kinetics, whereby slightly different time scales to attain steady state were observed for the different gas species. These differences are attributed to differences in (i) the species-specific diffusion kinetics in the membrane material and (ii) the species-specific Henry coefficients controlling the partial-pressure gradients driving the diffusive fluxes of the gas species across the membrane (Luis, 2018). Nevertheless, we conclude that the partial pressures of all gas species attained steady

state after approximately 1 d.

[Figure 2 about here.]

The steady-state partial pressures of $N_2$, $O_2$ and Ar observed in the SOMM gas volume are 30 %–35 % lower than those in the LIMM, where the partial pressures are at solubility equilibrium with the coresponding dissolved-gas concentrations in the water. The gas depletion in the SOMM module is due to the slow gas/water exchange, which is insufficient to maintain a

solubility equilibrium while gas is consumed by the miniRUEDI at a 50 % duty cycle. The gas/water partitioning in the SOMM therefore operates at a dynamic equilibrium between the gas transfer across the membrane and the gas consumption by the miniRUEDI. Also, note the slight fractionation of the different gas species during the dynamic steady-state, which is a result



of the species-specific rates of gas transfer across the SOMM membrane. Overall, we conclude that with a gas-sampling duty cycle of 50 % the SOMM does not attain Henrys Law equilibrium, as it would be required for the GE-MIMS method.

## 3.2 Experiment II

The characteristic time for the gas/water exchange to attain steady state in the SOMM is approximately $T_0 \approx 1\,\mathrm{d}$ (see Experiment I). The time resolution for repeated dissolved-gas analysis with the SOMM, therefore, cannot be higher than 1 measurement per day. Hence, with a miniRUEDI gas sampling time of $T_s \approx 6.5\,\mathrm{min}$, the duty cycle for gas sampling from the SOMM can be as low as $T_s/T_0 \approx 0.5\,\%$. This much lower duty cycle would result in approximately $100\times$ less gas consumption from the SOMM than with the 50 % duty cycle of Experiment I. The gas depletion in the SOMM gas volume (30–35 % in Experiment I) would be reduced by a similar factor, and would therefore be expected to be less than the analytical precision of the miniRUEDI (1–5 % RSD typical).

To test the SOMM with the GE-MIMS protocol at a duty cycle of 0.5 %, we increased the intervals for gas sampling from the SOMM to 24 h, keeping the sampling time at 6.5 min. During the remainig 99.5 % of the time, the miniRUEDI sampled the gas from the LIMM headspace. Fig. 3 shows the time series of the partial pressures observed in the gas volumes of the SOMM and LIMM modules. The relative differences between the partial pressures observed in the two equilibrator modules range from 1–4.6 %, which is within the analytical uncertainty of the miniRUEDI analysis. The LIMM analysis shows a slight trend of the $O_2$ partial pressures in the groundwater throughout the test period, which is also reflected in the SOMM results. Overall, the results from the SOMM show no bias relative to the LIMM reference. The modified GE-MIMS protocol for the SOMM therefore yields accurate quantification of the dissolved gases in the water.

[Figure 3 about here.]

## 3.3 Field test

We tested the performance of a SOMM module with GE-MIMS analysis in a geological experiment at a borehole in the Mont Terri Underground Rock Laboratory (Switzerland). The CS experiment is designed to observe migration of $CO_2$-rich fluids along a fault and their interactions with the surrounding clay rock (Zappone et al., 2021). After an initial phase of monitoring the natural background conditions, fluids will be artificially injected into the fault at high pressure via an injection borehole. The transport and geochemical evolution of the fluids are analyzed at an observation borehole downstream of the injection point. The water in the observation borehole is circulated at in-situ pressure (8.8 bar) in a closed loop to/from the SOMM membrane module. The SOMM water volume and the circulation lines between the module and the observation borehole were



filled with water that initially contained dissolved gases at partial pressures close to atmospheric equilibrium. Three months after starting the water circulation between the observation borehole and the membrane module, a miniRUEDI instrument was connected to the SOMM to test GE-MIMS analysis of the water in the observation borehole as developed in Experiment II. The adapted SOMM GE-MIMS analysis operated fully autonomous during the field test.

Fig. 4 shows the time series of the He, $N_2$, $O_2$ and Ar partial pressues determined in the observation-borehole water at daily

intervals during a 4 week long test run. The data gaps in these time series are related to artifacts introduced by manual sampling of the water from the observation borehole. The He partial pressure is approximately two orders of magnitude higher than that in air-saturated water (ASW, see $p_{atm}$ values in Fig. 4), which is due to the accumulation of radiogenic He produced in the host rock. The He partial pressure increased throughout the test period, which indicates that He equilibration between the borehole and the surrounding host rock is still ongoing due to the slow transport of water and solutes within the rock. The $N_2$, $O_2$ and

Ar partial pressures do not show any trends throughout the test period and seem to show only small variations (e.g., after the manual water sampling) that are within the analytical uncertainty of the miniRUEDI. The $N_2$ and Ar partial pressures show a slight excess relative to ASW, reflecting the typical excess-air signature in groundwater (see Kipfer et al., 2002; Aeschbach-Hertig and Solomon, 2013, for reviews). The low $O_2$ partial pressure in the observation-borehole water reflects the anoxic conditions in the pore water surrounding the observation borehole. However, the water in the observation borehole is not fully

anoxic, which, similar to He, is attributed to the incomplete equilibration of the initially air-loaded sample water with the anoxic pore water surrounding the observation borehole. Overall, the partial pressures observed with the adapted SOMM GE-MIMS technique correspond to the expected dissolved-gas partial pressures of the water in the observation borehole.

[Figure 4 about here.]

## 4 Discussion and conclusions

We experimentally determined the dynamics of the gas exchange between the water and the gas volumes in the SOMM membrane module. As expected, the mechanically robust construction of the SOMM membrane elements entails substantially lower gas-exchange rates than those achieved with the microporous membrane elements in the LIMM modules commonly used for GE-MIMS. Our tests indicated a characteristic time scale for attaining steady-state gas/water partitioning between the water and the gas volumes in the SOMM module of approximately 1 d, which is about $100\times$ slower than with the LIMM. Also, if

the SOMM module is used with the standard protocol for GE-MIMS analysis, we observed a depletion and an elemental fractionation of the gases in the SOMM because the gas transfer across the membrane in the SOMM is insufficient to maintain the





gas/water partitioning at solubility equilibrium. The SOMM is therefore not suitable for accurate dissolved-gas quantification using the standard GE-MIMS protocol.

We therefore developed a GE-MIMS protocol that is adapted to the characteristics of the gas exchange dynamics of the SOMM module. Given the slow gas exchange in the SOMM, the time resolution of the GE-MIMS measurements cannot be higher than 1 measurement per day. In contrast to the standard GE-MIMS protocol with the LIMM, which allows approximately 100 measurements per day at a 100 % gas-sampling duty cycle, there is therefore no need for continuous gas sampling from the SOMM to the gas analyzer. This allows using a very low duty cycle for gas sampling to substantially reduce the gas consumption from the SOMM module with the adapted GE-MIMS protocol. The low duty cycle avoids the gas depletion in the SOMM and thereby allows the gas/water partitioning in the the SOMM module to attain an equilibrium that is controlled only by the specific solubilities of the involved gas species, as required for the GE-MIMS technique. Dissolved-gas measurements from this adapted SOMM GE-MIMS technique showed no bias or extended scatter relative to reference measurements from the standard GE-MIMS protocol using a LIMM. The adapted SOMM GE-MIMS technique therefore allows accurate and precise quantification of dissolved gases. The analytical time resolution that can be achieved with this adapted technique is limited by the SOMM gas/water equilibration time of approximately 1d. For time-series measurements at shorter time intervals, the gas equilibrator module would need to be modified to allow quicker gas/water equilibration and possibly also to provide higher gas/water mass transfer across the membrane to balance the increased gas consumption by the gas analyser.

We demonstrated the analytical potential of the adapted SOMM GE-MIMS technique in a field test at a deep geological borehole containing water at an in-situ pressure of 8.8 bar. The adapted SOMM GE-MIMS technique allowed simple and autonomous quantification of dissolved gases, and therefore overcomes the limitations of previous attempts to apply the GE-MIMS technique in high-pressure water using complex depressurisation techniques. The SOMM therefore opens new opportunities for efficient and reliable on-site quantification of dissolved gases in water at high pressures, such as in deep geological systems. While our field demonstration was related to an experiment targeting the migration and fate of injected $CO_2$ and other dissolved gases in the context of $CO_2$ sequestration, the adapted SOMM GE-MIMS technique will also be most suitable for applications in research and monitoring of underground storage of other waste deposits, as well as in the exploration of water, fossil fuels, heat, hydrogen, helium, and other natural resources.

*Author contributions.* Conceptualization: MB, AR, AZ, RK. Methodology, Data analysis: MB, AR, RK. Investigation, project administration, : MB, AR. Resources: MB, AR, JG, AZ. Writing, visualization: MB. Review and editing: MB, AR, JG, AZ, RK.



*Competing interests.* There are no competing interests.



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





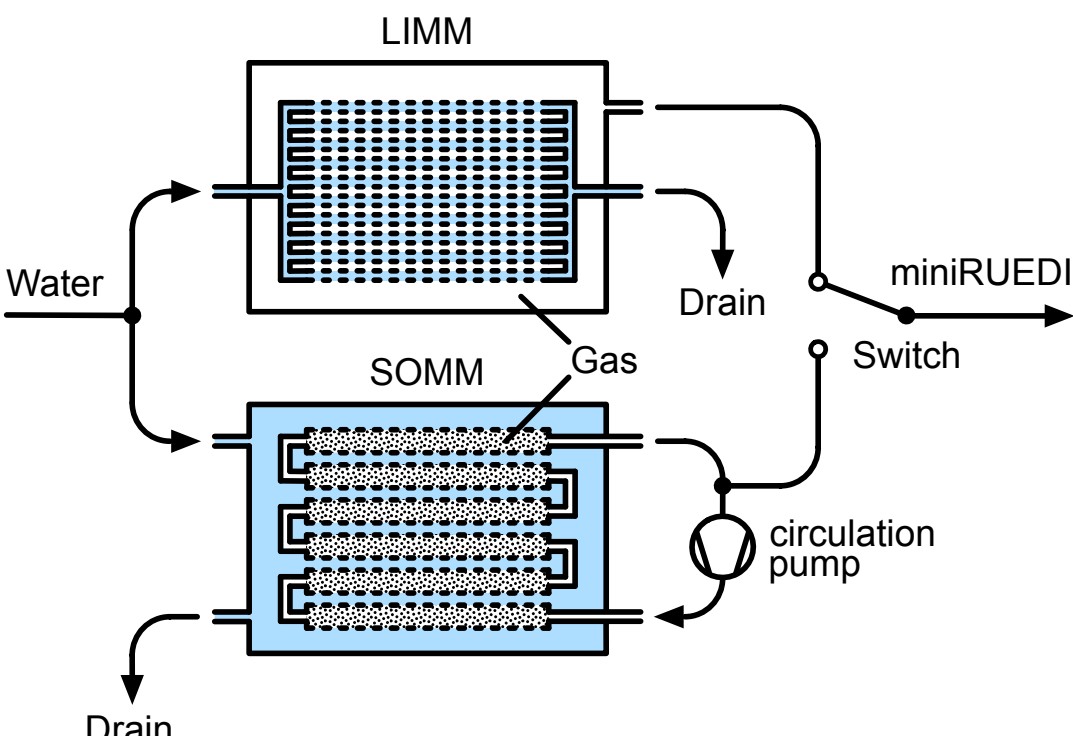

**Figure 1.** Experimental setup to test and compare the LIMM and SOMM membrane modules. Note the reversed gas/water arrangement on the lumen and shell sides of the membrane elements in the LIMM and SOMM.



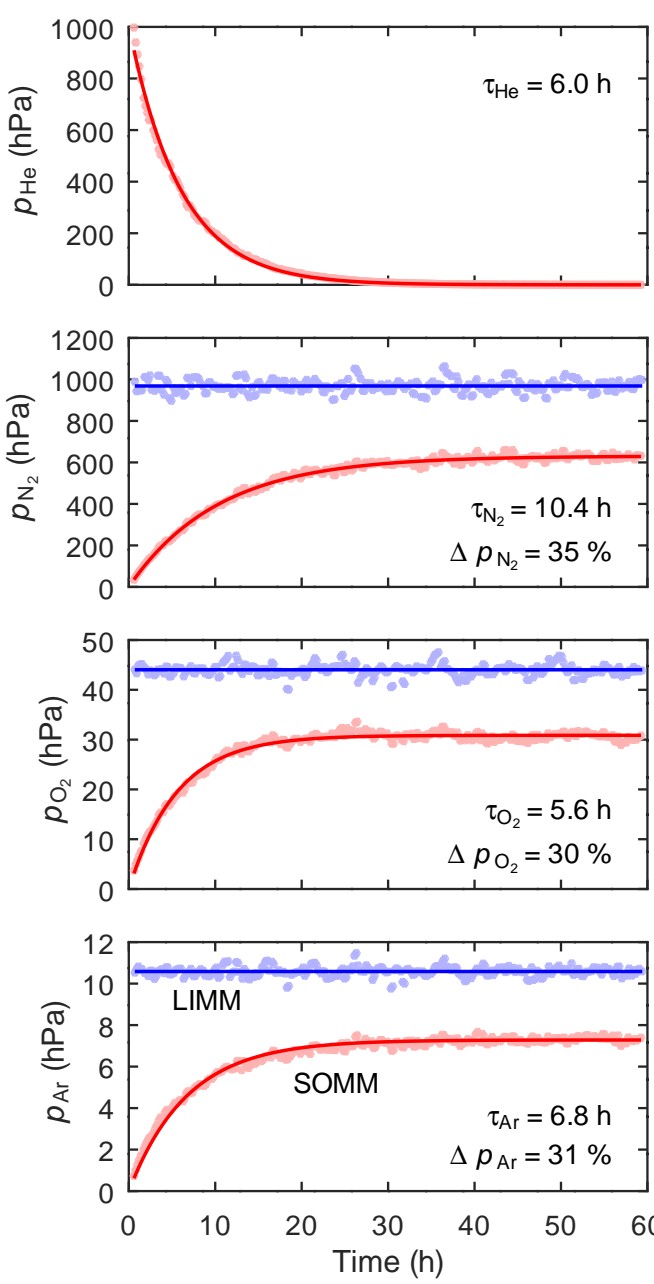

**Figure 2.** Experiment I: Evolution of the partial pressures of He, $N_2$, $O_2$ and Ar in the SOMM module, compared to those in the LIMM module. The curved lines correspond to a function of the form $p_{He} = p_{He}^* \times e^{-t/\tau_{He}}$ or $p_i = p_i^* \times (1 - e^{-t/\tau_i})$ for $i = N_2, O_2, Ar$; where $t$ is time and $p_i^*$ and $\tau_i$ are the gas-specific curve-fit parameters. The $\Delta p_i$ indicate the depletion of the dynamic steady-state partial pressures in the SOMM ($p_i^*$) relative to the equilibrium partial pressures in the LIMM.

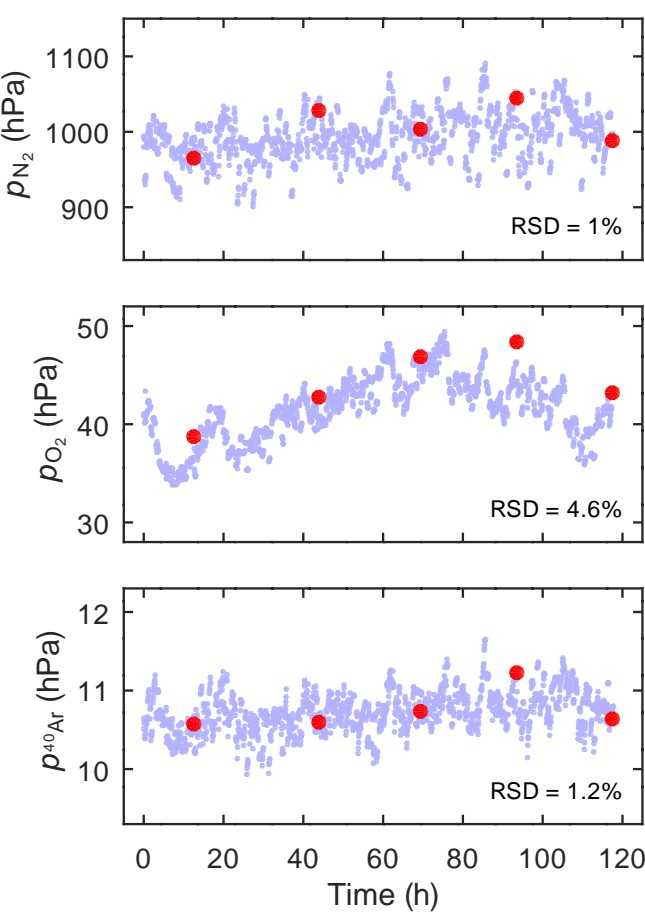

**Figure 3.** Experiment II: Comparison of the $N_2$, $O_2$ and Ar partial pressures determined with GE-MIMS using the LIMM as a reference (small blue dots) and the SOMM with a reduced sampling duty cycle of 0.5 % (large red dots). The RSD values indicate the relative standard deviations of the SOMM data relative to those of the LIMM.



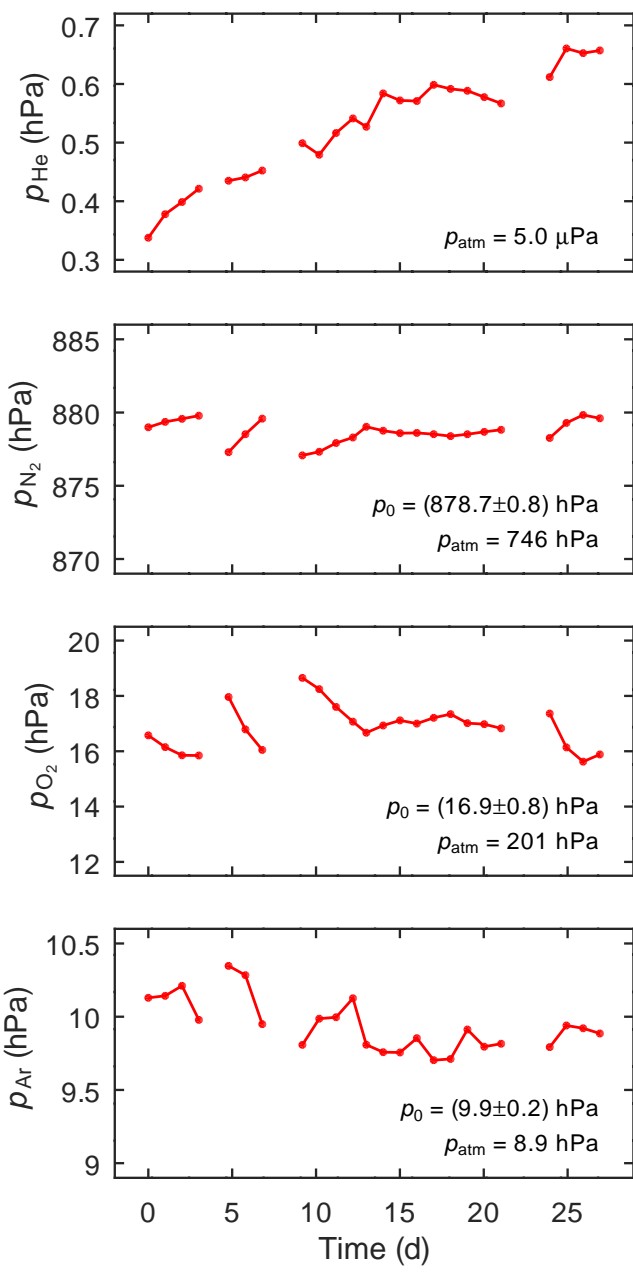

**Figure 4.** Partial pressures of He, $N_2$, $O_2$ and Ar dissolved in the observation-borehole water analysed in the SOMM GE-MIMS fieldtest. $p_0$ indicates the mean and standard deviation of the partial pressure values observed in the time series. $p_{atm}$ indicates the partial pressure of the respective gas in the atmosphere at the elevation of the study site. See text for explanation of data gaps.



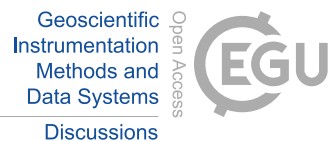

**Table 1.** Comparison of LIMM (GE-MIMS standard) and SOMM specifications.

|  | LIMM | SOMM |
| --- | --- | --- |
| Max. liquid/gas pressure difference | $2.1 - 4.1$ bar (a) | 100 bar |
| Membrane material | Microporous polypropylene | Non-porous silicon based polymer |
| Membrane thickness ($W$) | 0.03 mm | 1.6 mm |
| Gas volume ($V$) | 78 cm$^3$ | 40 cm$^3$ |
| Effective membrane area ($A$) | 58 dm$^2$ | 2 dm$^2$ |
| $V/A$ ratio | 1.3 cm$^3$/cm$^2$ | 40 cm$^3$/cm$^2$ |

(a) Depending on liquid temperature