# Peer review of "Gas-Equilibrium Membrane Inlet Mass Spectrometery (GE-MIMS) for water at high pressure"

_Geoscientific Instrumentation, Methods and Data Systems, 2023_

## Author Comment (AC1)

**Revision of GI manuscript gi-2023-12**

November 7, 2023

We carefully revised our manuscript "Gas-Equilibrium Membrane Inlet Mass Spectrometery (GE-MIMS) for water at high pressure" (gi-2023-12) according to the comments by the two reviewers. We thank the reviewers for their helpful comments, and we added an acknowledgment to the manuscript.

Below, you will find a detailed list of the review comments with our with our replies and descriptions of how we improved the manuscript. For consistency with the reviewer comments, line numbers given in the revision notes correspond to the first submission of the manuscript (the line numbers of the revised manuscript are sometimes slightly different). We provided a version of the revised manuscript where all changes are marked by colored text.

The comments by the two reviewers allowed us to further improve our manuscript. We hope the revised manuscript is now suitable for publication in GI.

Kind regards
Matthias Brennwald

**1 RC1: Anonymous Referee 1, 29 Aug 2023**

**Overview**

In this manuscript, the authors present the results of a new technique for dissolved gas measurements in subsurface water samples at high pressures. The technique is based on gas-liquid equilibration (GE-MIMS device), and measurement with a residual gas analyzer (the mini RUEDI). The mini RUEDI coupled to GE-MIMS was shown to perform well for the determination of dissolved gas, but usually uses a LIMM membrane that breaks at water pressures above approximatively 3 bar. In this paper, the authors describe tests made with a SOMM membrane that can withstand water pressures up to 100 bar. The SOMM membrane was developed for this application. As highlighted by the authors, this work is of great relevance for the field as measurement of dissolved gas in high pressure fluids (e.g. deep subsurface) is up to now still a technical challenge.

The paper is well structured, and the results give proof of evidence for the authors' statements. The paper should be accepted, with minor technical precisions. Please find below some questions and minor comments for the improvement of the manuscript.

**General remarks**

1. **Comment:**
   I would appreciate in the discussion an opening on challenges that could occur at higher pressures (i.e. closer to the maximum that can withstand). Does the equilibration time with the SOMM membrane (and thus the required maximum duty cycle) vary with the water pressure?

   **Reply:**
   The water pressure corresponds to the mechanical force excerted by the water onto the confining surfaces inside the membrane module. This static mechanical force has no relation to the concentration of dissolved gases in the water or to the properties and processes related to the gas exchange in the SOMM. There are therefore are no grounds for a discussion of such effects.

   **Changes to the manuscript:**
   No change.

2. **Comment:**
   Could suspended solids be a challenge for the use of such membranes (LIMM and SOMM)? In the different experiments was the water filtered to avoid such issue?

   **Reply:**
   The experience with the LIMM is that suspended solids can be a challenge, depending on the application and environment where the LIMM is used. If the water contains suspended particles, they tend to clog the fine membrane tubing of the LIMM and thereby reduce the water flow. The established solution to avoid clogging of the LIMM is to insert a particle filter to the water flow upstream of the LIMM. If clogging turns out to also be relevant with the SOMM in specific applications, the solution would be to use a particle filter in the same way as with the LIMM.

However, the scope of this work was to test the gas/water exchange dynamics within the SOMM, and to develop an adapted analytical protocol to allow accurate GE-MIMS analyis with the SOMM. The potential of the SOMM for clogging by suspended particles is not in this scope and was therefore not tested.

**Changes to the manuscript:**
No change.

3. **Comment:**
What are the current methods used for measurement of dissolved gases in the subsurface (other than GE-MIMS coupled to mini RUEDI)? What are the advantages of the mini RUEDI compared to these methods? This could be a nice addition to the introduction/discussion.

**Reply:**
A number of references to other methods is given on lines 27–29. The advantages of the miniRUEDI/GE-MIMS technique are described on lines 29–46.

**Changes to the manuscript:**
No change.

**Specific comments**

4. **Comment:**
Line 53: typo, However

**Reply:**
Agreed.

**Changes to the manuscript:**
Changed "howeber" to "however".

5. **Comment:**
Lines 93-94: please give information on the local aquifer in term of location, turbidity, suspended solids (see general remark 3)

**Reply:**
See also comment 21. The water was taken from a coninuously pumped well

in a riparian aquifer (where suspended particles would be filtered by the aquifer matrix).

**Changes to the manuscript:**
Line 93–94: Added details and reference to aquifer description to the manuscript.

6. **Comment:**
Lines 93-94: please precise the parameters of the mini RUEDI used (i.e. flushing time, number of repetition on the detectors)

**Reply:**
This paragraph is related to the experimental setup shown in Fig. 1, not to the configuration of the mass-spectrometric parameters used with the miniRUEDI analyser. However, we agree that additional detail on the miniRUEDI configuration would be useful, see comment 22.

**Changes to the manuscript:**
See comment 22.

7. **Comment:**
Line 139: "SOMM can be as low as" change to "SOMM should be as low as"

**Reply:**
It seems the reviewer was confused by the previous sentence, which was poorly phrased.

**Changes to the manuscript:**
Line 137: Changed "cannot" to "does not need to".

8. **Comment:**
Line 147: "slight trend", please precise (i.e. Increase? Decrease? Significative?)

**Reply:**
The O2 partial pressure shows a slight increase during the first 75 hours, then decreases again.

**Changes to the manuscript:**
Changed "trend" to "variation".

9. **Comment:**
Lines 153-163: should go to material and methods (as a 2.5 Field test subchapter)

**Reply:**
Section 3.3 is meant as an add-on demonstration of the use of the new SOMM method in a real-world application. It is not meant as part of the method design and development process.

**Changes to the manuscript:**
Lines 152–153: Changed title and rephrased first sentence of Section 3.3 to clarify that this section is not part of the method design/development.

10. **Comment:**
Line 189: is the GE-MIMS protocol only about changing the duty cycle, or were other parameters of the mini RUEDI changed as well?

**Reply:**
Yes, it's only about adapting the duty cycle, as described in the manuscript (for example on lines 197-199).

**Changes to the manuscript:**
No change.

11. **Comment:**
Line 192: "there is therefore no need for . . . ", what do you mean by no need? The conclusion should be that with SOMM the temporal resolution cannot be as high as with LIMM. Using the term "no need" sounds like it is a default of LIMM that we can sample more often.

**Reply:**
See also comments 12 and 13. We agree that the phrasing of lines 191–194 was a bit convoluted.

**Changes to the manuscript:**
Lines 192–194: Revised the text to use shorter sentences with a better logical continuity.
Also included suggestions from comments 12 and 13..

12. **Comment:**
Line 193: "allows", same comment as 8̶11: this is not really a positive aspect, should be replaced by something like "the very low duty cycle is also required to prevent gas depletion in the gas phase, and the resulting offset of the partial pressures measured with the mini RUEDI"

**Reply:**
See comment 11.

**Changes to the manuscript:**
See comment 11.

13. **Comment:**
Line 194: "avoids", same comment as a 8̶11 and 9̶12: this is not really a positive aspect, should be replaced by "is necessary to"

**Reply:**
See comment 11.

**Changes to the manuscript:**
See comment 11.

14. **Comment:**
Lines 209-211: it would be nice to add references of studies where dissolved gas measurement was a limitation/was hindered

**Reply:**
Two references in this context were already discussed on line 53. A third paper was recently published.

**Changes to the manuscript:**
Added reference to Weber et al. 2023 on line 53 (and also on lines 46 and 155).

**Comments on figures**

15. **Comment:**
Figure 3: which LIMM data did you use for each SOMM data for the calculation of RSD, average of SOMM data/average of LIMM data?

**Reply:**
The LIMM reference value for comparison with the SOMM value was determined by intpolating the LIMM data to the times of the SOMM data points.

**Changes to the manuscript:**
Added this information ot the figure caption.

**2 RC2: Anonymous Referee 2, 17 Oct 2023**

**Overview**

This manuscript describes a useful extension for the on-site analysis of gases dissolved in water, in that higher pressure water can be analysed. The manuscript should be accepted for publication after a few clarifications are made, and some further practical details included in respect to the analysis parameters of the miniRUEDI, so that an interested researcher could reproduce the results.

16. **Comment:**
    Ultimately, this study would have benefitted by passing large volumes of air saturated water (or some other standardised water) through both the SOMM and the LIMM devices to demonstrate the accuracy of each - presumably this was deemed infeasible due to the large volumes of water required. The reader is left to trust in the excellent published work of the authors (especially MB and RK) that the LIMM-based GE-MIMS/miniRUEDI system provides accurate results within stated error margins, and therefore, because the LIMM results are consistent with the SOMM results then the SOMM system is also accurate. Some mention of the absolute accuracy of the method should be made, why a water standard was not measured, and some reference made to the logical sequence reliant on LIMM providing accurate gas partial pressures.

    **Reply:**
    The GE-MIMS method has previously been tested using different membrane modules (including the LIMM) (Brennwald et al., 2016, as cited in the manuscript). These tests included a validation using river water that was assumed to be in equilibrium with ambient air (air saturated water, ASW), and independent work has confirmed the accuracy and precision of the GE-MIMS technique (as described on lines 91–93). The existing GE-MIMS techniques therefore provides a robust reference for comparison with new techniques and membrane modules.

    The objective of this work was to test the utility of the SOMM module for GE-MIMS. The SOMM was confirmed to provide results that are consistent the with the LIMM if the duty cycle is adjusted as described in the manuscript. The air-saturated water (ASW) test therefore seems unnecessary. Also, as indicated by the reviewer, huge amounts of ASW would be required ($2\,\mathrm{m}^3$ or more for one single measurement), which would indeed be challenging to handle.

    The consistency of the SOMM measurements with those of the LIMM (used as a reference) is explicitly discussed in the manuscript (for example on lines 146–150). However, we agree that it would be useful to provide a more explicit reference to the previous GE-MIMS validation.

**Changes to the manuscript:**
Line 146: Added a reference to Brennwald et al. 2016.
Line 150: Added a statement that the analytical uncertainty with the SOMM is consistent with the that of the standard GE-MIMS/LIMM.

**Specific comments**

17. **Comment:**
Line 9,18,134: Henry's Law should have an apostrophe

    **Reply:**
Agreed. Also on line 35.

    **Changes to the manuscript:**
Added apostrophe on lines 9, 18, 35 and 134.

18. **Comment:**
Line 38: It is not immediately clear whether gas transfer across a membrane would fractionate a gas mixture, or if any such fractionation could have a dependence on the dissolved gas composition - this is doubly relevant for the present manuscript due to there being two different membrane materials being investigated. Please expand on whether gas species fractionation across a membrane is inherently obviated due to the design or how these GE-MIMS systems are used (e.g., adequate dead volume present; time allowed for differential equilibration across the membrane to become unimportant, etc.), and/or provide more specific citations for how this aspect has been demonstrated in peer-reviewed literature (Mächler 2012 is a PhD thesis, and might not qualify as peer-reviewed, and Brennwald et al. 2016 is similarly general as to the reasons no fractionation is present).

    **Reply:**
The point of the GE-MIMS technique is that it avoids fractionation of the gas species by principle, as summarised in the manuscript on lines 33–40 (in short: the GE-MIMS principle is based on gas/water partitioning according to Henrys Law. It is *not* based to the dynamics of gas diffusion across the membrane and thereby avoids any diffusive fractionation effects). Full details and validation on the GE-MIMS background and principles *are* given in Brennwald et al., 2016, as cited in the manuscript (for example on line 40).

    **Changes to the manuscript:**
No change.

19. **Comment:**
Line 53: However is misspelled

**Reply:**
See comment 4.

**Changes to the manuscript:**
See comment 4.

20. **Comment:**
Line 77: Is it possible to provide a link to this patent application, should a researcher require further details of the design and fabrication materials? If the SOMM is commercially available at present, please confirm this in response to reviews - if it is not available at present, state this and when it will be available in the manuscript.

**Reply:**
The patent application was received by the European patent office with the application nr. 2210240.2. The SOMM module is available from Solexperts AG in Switzerland.

**Changes to the manuscript:**
(1) Added patent application number on line 58.
(2) Removed "(patent pending)" on line 77 (because this is already stated on line 59).

21. **Comment:**
Line 93: Can the authors make a statement about the concentrations of the dissolved gases in this "local aquifer" (e.g., has it been measured by other techniques, such as gas chromatography or mass spectrometry)? If these concentrations are unknown and/or unnecessary to be stated for the hypothesis being tested this should be clearly noted, but it would still be better to give the reader at least a qualitative idea of the water being used in this test.

**Reply:**
See comment 5.

**Changes to the manuscript:**
See comment 5.

22. **Comment:**
Line 118: Was the data here (and in the rest of the manuscript) collected on a Faraday Cup, and using atmospheric pressure air as a normalising standard? Is the pressure at the inlet of the miniRUEDI monitored/recorded/used in the calculation for sample and standard? I recommend being specific on this in the text or in a supporting document, including whether the normalising gas was ambient air, and what considerations one might make about using ambient air in an underground lab as a standard.

**Reply:**
We agree that these details were missing in the manuscript (see also comment 6). These information are relevant for all lab tests in this work, not just Experiment 1. We therefore added the information to section 2.2.

**Changes to the manuscript:**
Added a paragraph with the miniRUEDI analytical details after line 100.

23. **Comment:**
Line 154: Define CS

    **Reply:**
    CS = Carbon Storage (this was missing in the manuscript)

    **Changes to the manuscript:**
    Added this information on line 154.

24. **Comment:**
Line 163: I think "autonomous" (adjective) should be "autonomously" (adverb).

    **Reply:**
    Agreed.

    **Changes to the manuscript:**
    Changed as suggested.

25. **Comment:**
Line 167: I am not sure the authors have correctly defined "(ASW, see patm values in Fig. 4)". The N2/Ar ratio in air is 84; the ratio in Figure 4 is also 84. In contrast, the N2/Ar ratio in air saturated water is 38 (at sea level, 20 degrees C). I suspect what the authors actually mean is that Patm is the partial pressure in the membrane equilibrator headspace when air saturated water is passed through the equilibrator, producing an exsolved equilibrium gas mixture which is equivalent to air (unless there are any artefacts to the membrane system, which the authors imply are negligible). I feel it would be helpful for the authors to make this somewhat subtle but ultimately confusing point more clear in the text and the Figure legend.

    **Reply:**
    It seems this sentence was confusing to the reviewer due to the ASW reference (see also comment 26). The ASW reference is not necessary here, so we removed it to avoid any confusion.

    **Changes to the manuscript:**
    Removed the confusing "ASW" abbreviation.

26. **Comment:**

Lines 171-172: The phrase "The N2 and Ar partial pressures show a slight excess relative to ASW" also suffers from the logical discontinuity between dissolved gas concentrations in ASW and those measured in ASW passed through a membrane equilibrator and measured on a miniRUEDI. It would be helpful to be more specific, to avoid confusion.

**Reply:**

The text refers only to the *partial pressures* of the gases dissolved in the water, while the reviewer seemingly misread this as the *concentrations* of dissolved gases in water, possibly as a follow up from comment 25.

**Changes to the manuscript:**

Line 172: Replaced "relative to ASW" to "relative to those in air-saturated water", where *those* explicitly refers to the *partial pressures* of the preceding phrase.

27. **Comment:**

Figure 4: The notation on the He panel is confusing. For all panels "Patm = xx hPa" is shown, except for the He panel which appears to be alphaPa, which I presume to be a typo of some kind.

**Reply:**

$p_{atm}$ is given in $\mu$Pa, not "alphaPa". It seems the PDF viewer or the printer used by the reviewer did not interpret this correctly.

**Changes to the manuscript:**

No change.